

# Relation extraction in Chinese using attention-based bidirectional long short-term memory networks

Yanzi Zhang

College of Chinese Language and Culture, Jinan University, Guangzhou, China

## ABSTRACT

Relation extraction is an important topic in information extraction, as it is used to create large-scale knowledge graphs for a variety of downstream applications. Its goal is to find and extract semantic links between entity pairs in natural language sentences. Deep learning has substantially advanced neural relation extraction, allowing for the autonomous learning of semantic features. We offer an effective Chinese relation extraction model that uses bidirectional LSTM (Bi-LSTM) and an attention mechanism to extract crucial semantic information from phrases without relying on domain knowledge from lexical resources or language systems in this study. The attention mechanism included into the Bi-LSTM network allows for automatic focus on key words. Two benchmark datasets were used to create and test our models: Chinese SanWen and FinRE. The experimental results show that the SanWen dataset model outperforms the FinRE dataset model, with area under the receiver operating characteristic curve values of 0.70 and 0.50, respectively. The models trained on the SanWen and FinRE datasets achieve values of 0.44 and 0.19, respectively, for the area under the precision-recall curve. In addition, the results of repeated modeling experiments indicated that our proposed method was robust and reproducible.

## INTRODUCTION

Relation extraction (RE), one of the key topics in the domain of information extraction (IE), focuses on extracting semantic relationships that exist between pairs of entities within natural language sentences (*Konstantinova, 2014*). This application plays an important role in various downstream applications to develop large-scale knowledge graphs. The recent growth of deep learning has significantly powered neural relation extractions (NRE), wherein the utilization of deep neural networks enables the automatic learning efficiency of semantic features (*Zhou et al., 2016*; *Lin et al., 2016*; *Jiang et al., 2016*; *Zeng et al., 2014*; *Liu et al., 2013*). While feature engineering is not required in the context of neural relation extraction (NRE), it overlooks the crucial consideration that the diverse language granularity of the input has a notable influence on the model's performance, particularly in the case of Chinese relation extraction (RE). Most existing methods for Chinese RE can be categorized into two types based on the discrepancy in granularity: character-based RE and word-based RE (*Li et al., 2019*). The character-based RE model

Corresponding author
Yanzi Zhang,
qingwuyuer1986@163.com

treats each input sentence as a character sequence, but it can only capture a small number of features due to being restricted to fully using word-level information (*Qi et al., 2014*). For the word-based model, word segmentation is usually applied before fetching created word sequences through neural networks (*Geng et al., 2020*). Segmentation efficiency, therefore, may have a substantial influence on word-based models. Although both approaches have pros and cons, they share a common shortcoming in the extraction of sufficient semantic information (*Li et al., 2019*). The aforementioned limitation poses a challenge, particularly when working with sparsely annotated datasets. As a result, there is an urgent need for a more efficient approach capable of extracting copious semantic information from plain texts in order to reveal high-level entity relationships.

Recent years have seen a lot of research in natural language processing (NLP) regarding RE, especially NRE. *Liu et al. (2013)* first proposed a simple RE model using a convolutional neural network (CNN). Their work is considered an essential basis for automatic feature learning in later modern RE models. *Zeng et al. (2014)* developed another CNN model to first express the positional information with positional embeddings. The PCNNs model was subsequently introduced to establish the multi-instance learning paradigm for relation extraction. The problem of sentence selection, however, also affects the PCNNs model (*Zeng et al., 2015*). To improve it, *Lin et al. (2016)* employed the attention technique to every instance in the bag. Another model adapting to multi-instance and multi-label paradigms was also suggested to solve the problem (*Jiang et al., 2016*). Although PCNNs models are fairly robust, they are unable to use contextual information and it can only be solved with recurrent neural network (RNN) models. Hence, long-short-term memory (LSTM) incorporated with the attention mechanism was also used to handle the RE tasks (*Li et al., 2019*; *Zhou et al., 2016*; *Zhang & Wang, 2015*).

In the field of Chinese relation extraction, most approaches commonly utilize word- or character-based NRE models (*Xu et al., 2017*; *Rönnqvist, Schenk & Chiarcos, 2017*; *Zhang, Chen & Liu, 2017*; *Chen & Hsu, 2016*). These methods, however, concentrate on enhancing the model's performance while neglecting the reality that different granularities of input would have a great impact on RE models. Since character-based models are unable to use words, they extract fewer features compared to word-based ones (*Zhang & Yang, 2018*). Despite attempts by several approaches to incorporate word-level and character-level information in various NLP tasks, such as utilizing character bigrams (*Chen et al., 2015*; *Yang, Zhang & Dong, 2017*) and soft words (*Peng & Dredze, 2016*; *Chen, Zheng & Zhang, 2014*; *Zhao & Kit, 2008*), the efficiency of information extraction from these methods remains low. To deal with this problem, tree-structured RNN models were suggested. *Tai, Socher & Manning (2015)* developed a tree-like LSTM to enhance semantic representation. Human action recognition (*Sun et al., 2017*), neural network encoders (*Su et al., 2016*), NRE (*Zhang & Yang, 2018*), voice tokenization (*Sperber et al., 2017*), representation learning for multimedia (*Liu et al., 2022*; *Wang et al., 2022, 2023*), and recommender systems (*Li et al., 2023*) have all used this structure. The lattice LSTM models were demonstrated to improve the exploitation of word and word sequence information, but they failed to deal with the ambiguity of polysemy. The integration of external linguistic knowledge was used to partially solve this issue (*Shen et al., 2021*; *Liu et al., 2023*). *Li et al.*

**Table 1 Data for model development and evaluation.**

| Dataset | Number of samples (relations) | | |
|---------|----------|------------|------|
| | Training | Validation | Test |
| SanWen | 17,227 | 1,793 | 2,220 |
| FinRE | 13,306 | 1,455 | 3,653 |

*(2019)* proposed a model using multi-grained information and external linguistic knowledge to perform Chinese RE tasks. Their models used sense-level information supported by HowNet, a concept knowledge base containing Chinese annotation and correlative word senses. Although these approaches achieve satisfactory outcomes, there is a lot of room for model improvement (*Dong et al., 2022*; *Lu et al., 2023*).

This study introduces a more effective model for Chinese RE tasks by utilizing bidirectional LSTM (Bi-LSTM) along with an attention mechanism. The goal is to extract the most crucial semantic information from a sentence. Unlike existing models, our approach does not rely on any features derived from lexical resources or NLP systems as external domain knowledge. The main contribution of this article lies in applying the attention mechanism within the Bi-LSTM network, enabling automatic focus on the decisive words. By doing so, our model eliminates the need for external knowledge and NLP systems in the development of RE models.

## MATERIALS AND METHODS

### Datasets

We conducted our modeling experiments on two benchmark datasets: Chinese SanWen and FinRE. The Chinese SanWen dataset comprises 837 Chinese literature articles, encompassing nine categories of relations. Among these articles, 695 were used for training, 58 for validation, and the remaining 84 for testing. On the other hand, the FinRE dataset was constructed from a source of 2,647 financial news articles from Sina Finance. It consists of 13,486 relations for training, 1,489 for validation, and 3,727 for testing. The FinRE dataset encompasses 44 distinct relationships, including a special '*NA*' relation indicating an unrelated entity pair. These datasets were collected from *Li et al. (2019)*'s study. Table 1 gives detailed information on the number of samples (relations) in the training, validation, and test sets of each dataset.

### Word embeddings

For a sentence that builds of $T$ words $S = \{x_1, x_2 \ldots, x_T\}$, each word $x_i$ will be expressed as a numeric vector $e_i$. For every single word in $S$, its embedding matrix $W^{wrd} \in \mathbb{R}^{d^w} |V|$ is first looked up, where $d^w$ is the word embedding size and $V$ is a fixed-sized vocabulary. Matrix $W^{wrd}$ is trainable, while $d^w$ is a hyper-parameter defined by users. The word $x_i$ is then converted into its corresponding word embedding matrix $e_i$:

$$e_i = W^{wrd} v_i, \tag{1}$$

where $v_i$ is a vector with size of $|V|$, in which the index of $e_i$ is assigned 1 while the indices of other positions are assigned 0. Finally, the sentence, in the form of a numeric vector $embs = \{e_1, e_2, ..., e_T\}$, is passed through the next layer.

## Model architecture

The long short-term memory (LSTM), proposed by *Hochreiter & Schmidhuber (1997)*, is a specialized type of recurrent neural network that is designed to effectively work with sequential data. Original LSTM incorporates memory cells and a set of gates that control the flow of information, allowing them to retain important information over extended time periods and mitigate the vanishing gradient problem. The key idea of LSTM is the introduction of the adaptive gating mechanism that decides which information will be retained or cleared. Various LSTM variants were created to address different problems. In our proposed method, we used the LSTM variant by *Graves, Rahman Mohamed & Hinton (2013)*, in which weighted peephole connections from the constant error carousel (CEC) are added to the same memory block's gates. This LSTM variant is designed to directly use the current cell state in creating the gate degrees. Hence, the peephole connections permit all gates to inspect the cell regardless of the status (closed/open) of the output gate.

This LSTM architecture is represented by sets of three weight matrices $W_{x_g}$, $W_{h_g}$, and $W_{c_g}$, and bias $b_g$, where $g$ is $i, f$, and $o$ for the 'Input' gate $i_t$, 'Forget' gate $f_t$, and 'Output' gate $o_t$, respectively. Each gate is programmed to calculate degrees by considering the current input ($x_i$), the previous state ($h_{i-1}$), and the peephole ($c_{i-1}$) of the cell. These factors determine which information should be stored or discarded, and the gate produces the output for the next state. The mechanism is described in the equations below:

$$u_t = \sigma(W_{c_u}c_{t-1} + W_{h_u}h_{t-1} + W_{x_u}x_t + b_u), \tag{2}$$

with $u = i$ and $u = f$, and

$$g_t = Tanh(W_{c_c}c_{t-1} + W_{h_c}h_{t-1} + W_{x_c}x_t + b_c), \tag{3}$$
$$c_t = f_t c_{t-1} + i_t g_t, \tag{4}$$
$$o_t = \sigma(W_{c_o}c_t + W_{h_o}h_{t-1} + W_{x_o}x_t + b_o), \tag{5}$$
$$h_t = Tanh(c_t)o_t. \tag{6}$$

Figure 1 visualizes the attention-based Bi-LSTM architecture used in our model. The previous cell state and current information returned from the cell are summed to obtain the current cell state $c_t$. For numerous sequence modeling tasks, it is advantageous to have access to both past and prospective contexts. However, traditional LSTM networks follow a sequential approach in processing sequences, neglecting potential contextual information. Bidirectional LSTM networks allow the learning process to proceed in both directions—forward and reverse—to fully exploit information from both past and prospective contexts. The output of the $i^{th}$ word is calculated as:

$$h_i = [\overrightarrow{h_{i\,forward}} \oplus \overrightarrow{h_{i\,backward}}]. \tag{7}$$

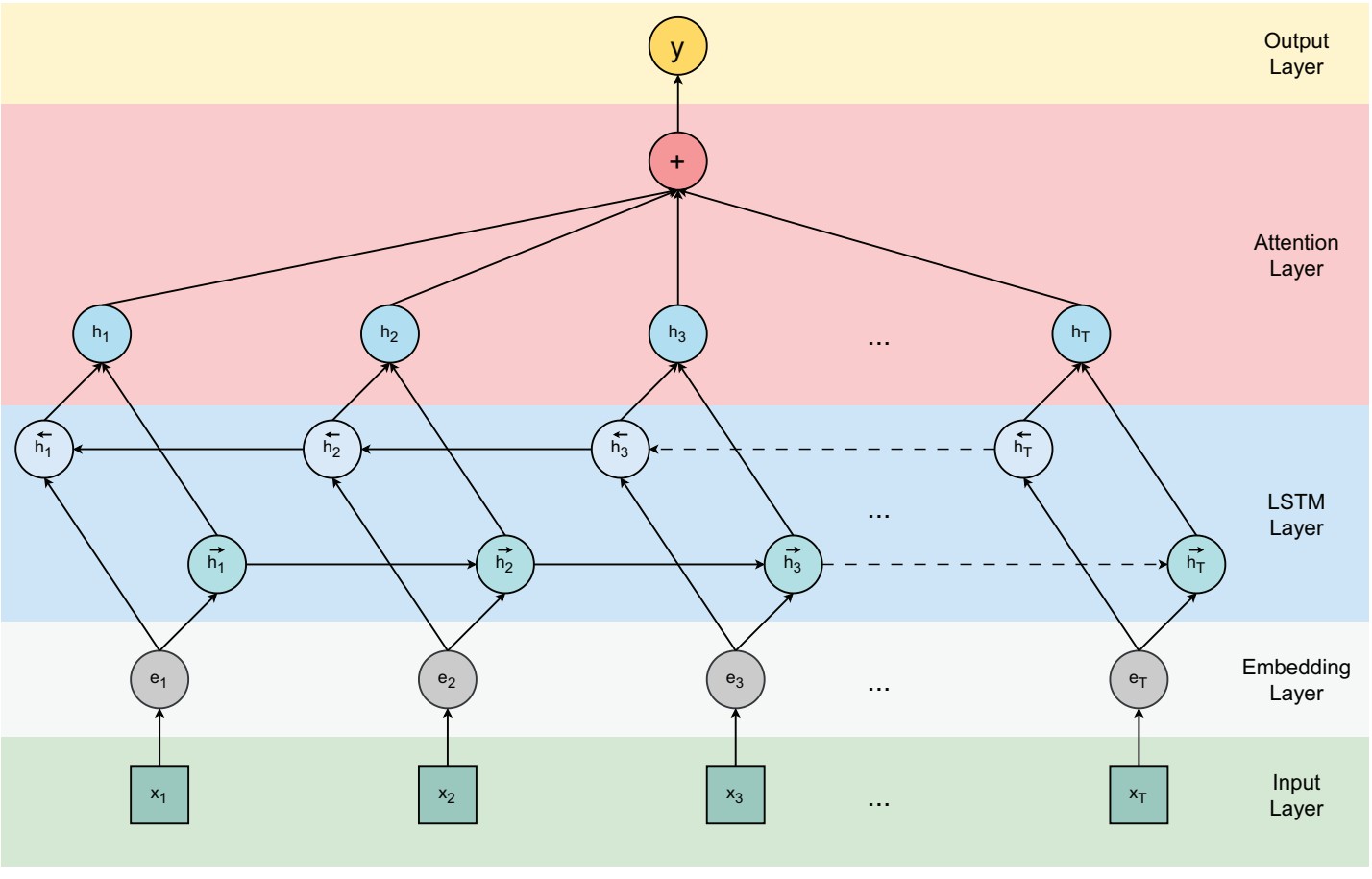

**Figure 1** Attention-based bidirectional LSTM model.

## Attention

The attention mechanism is a powerful technique in deep learning models that enables the network to focus on specific parts of the input sequence while performing computations. It allows the model to allocate varying degrees of importance to different elements of the input, enhancing its ability to capture relevant information. By dynamically weighing the importance of different parts of the input, the attention mechanism improves the model performance. Recently, attention-based neural networks have demonstrated remarkable achievements across various tasks, such as question answering (*Lu et al., 2016*), machine translation (*Luong, Pham & Manning, 2015*), speech recognition (*Chorowski et al., 2015*), and image captioning (*Huang et al., 2019*). In our research, we incorporated an attention mechanism to enhance the effectiveness of relation classification tasks. We constructed matrix $H$, comprising of output vectors generated by the LSTM layer $[h_1, h_2, ..., h_T]$, where $T$ represents the length of the sentence. The sentence representation $r$ is then derived by computing a weighted summation of these output vectors:

$$\alpha = Softmax(w^T Tanh(H)), \tag{8}$$

$$r = H\alpha^T, \tag{9}$$

where $H \in R^{d^w \times T}$ in which $d^w$ is the dimension of the word vectors and $w$ is a trained parameter vector. $w$, $\alpha$, and $r$ have dimensions of $d^w$, T, and $d^w$, respectively. The final representation of the sentence pair is obtained for classification as follows:

$$h^* = Tanh(r). \tag{10}$$

## Classification

In order to carry out the classification task, a softmax classifier was employed in predicting label $\hat{y}$ from set of classes $Y$ for a given sentence $S$. The input for the classifier is hidden state $h^*$:

$$\hat{p}(Y|S) = Softmax(b^{(S)} + W^{(S)}h^*), \tag{11}$$

$$r = \arg\max_y \hat{p}(Y|S). \tag{12}$$

The loss function was the negative log-likelihood:

$$J(\theta) = -\frac{1}{m}\sum_{i=1}^{m} t_i log(y_i) + \lambda||\theta||_F^2, \tag{13}$$

where $m$ is the number of target classes, $\hat{y}$ is the log-likelihood of a true class label, $y \in \mathbb{R}^m$ is the predicted probability, $t \in \mathbb{R}^m$ is the one-hot encoding of the ground truth, and $\lambda$ is an L2 regularization hyperparameter. The L2 regularization was employed alongside the dropout technique to avoid overfitting.

## Regularization

To prevent co-adaptation among hidden units, the dropout technique was implemented, which involves randomly removing a certain number of nodes from the network during forward propagation (*Hinton et al., 2012*). The dropout was applied to three layers: the embedding layer, the LSTM layer, and the penultimate layer. Besides, the L2-norms of the weight vectors were constrained by rescaling $w$ to obtain $||w|| = s$ once $||w|| > s$ after gradient descent steps.

## Experiments

The models trained with the SanWen and FinRE datasets required 5.29 and 4.37 s for the training, respectively. PyTorch 1.3.1 was used as the training platform for these two models. The computing configuration for modeling tasks is specified by a workstation with 16 GB of RAM and one NVIDIA RTX-3060. Both models were trained over 100 epochs, and the training processes were terminated once validation losses were at a minimum. The prediction threshold of 0.5 was chosen as the default threshold. In our modeling experiments, the models for the SanWen and FinRE datasets converged at epochs 71 and 63, respectively.

**Table 2 The performance of our models on the test sets.**

| Dataset | Metric | | | |
|---|---|---|---|---|
| | AUCROC | AUCPR | ACC | F1 |
| SanWen | 0.6960 | 0.4389 | 0.4990 | 0.4336 |
| FinRE | 0.4970 | 0.1871 | 0.2697 | 0.1146 |

**Table 3 The performance of the models in the repeated modeling experiments using the SanWen dataset.**

| Trial | Metric | | | |
|---|---|---|---|---|
| | AUCROC | AUCPR | ACC | F1 |
| 1 | 0.7162 | 0.4398 | 0.5059 | 0.4135 |
| 2 | 0.7141 | 0.4195 | 0.4937 | 0.4146 |
| 3 | 0.6823 | 0.4082 | 0.4937 | 0.4044 |
| 4 | 0.7285 | 0.4335 | 0.5100 | 0.4245 |
| 5 | 0.7051 | 0.4282 | 0.5018 | 0.4156 |
| 6 | 0.6407 | 0.4082 | 0.4660 | 0.3291 |
| 7 | 0.7250 | 0.3976 | 0.5163 | 0.4215 |
| 8 | 0.6981 | 0.3991 | 0.4909 | 0.3579 |
| 9 | 0.7282 | 0.4806 | 0.5217 | 0.4365 |
| 10 | 0.7397 | 0.4810 | 0.5249 | 0.4529 |
| Mean | 0.7078 | 0.4296 | 0.5025 | 0.4070 |
| SD | 0.0289 | 0.0304 | 0.0175 | 0.0368 |
| 95% CI | [0.6899–0.7257] | [0.4107–0.4484] | [0.4917–0.5133] | [0.3842–0.4298] |

## RESULTS AND DISCUSSION

We conducted a comprehensive evaluation of our model using various performance metrics, including the area under the receiver operating characteristic curve (AUCROC), the area under the precision-recall curve (AUCPR), accuracy (ACC), and the F1 score (F1). The test results, as summarized in Table 2, provide insights into the performance of both models on their respective test sets.

The experimental findings indicate that the model developed using the SanWen dataset exhibits superior performance compared to the model developed with the FinRE dataset. Specifically, the SanWen-based model achieves an AUCROC value of 0.6960 and an AUCPR value of 0.4389, showcasing its ability to effectively discriminate between positive and negative instances. The FinRE-based model demonstrates lower values of AUCROC (0.4970) and AUCPR (0.1871), indicating a relatively weaker discriminatory power. Furthermore, the accuracy and F1 score align with the aforementioned pattern, highlighting the consistent superiority of the SanWen-based model over its FinRE-based counterpart.

Modeling experiments were conducted with a focus on assessing the stability of the proposed model for Chinese relation extraction. To ensure comprehensive evaluation, these experiments were repeated a total of 10 times (Tables 3 and 4) using different

**Table 4 The performance of the models in the repeated modeling experiments using the FinRE dataset.**

| Trial | Metric | | | |
|---|---|---|---|---|
| | AUCROC | AUCPR | ACC | F1 |
| 1 | 0.5095 | 0.1769 | 0.2673 | 0.1127 |
| 2 | 0.5032 | 0.1922 | 0.2688 | 0.1139 |
| 3 | 0.4979 | 0.1709 | 0.2635 | 0.1099 |
| 4 | 0.5140 | 0.1706 | 0.2642 | 0.1104 |
| 5 | 0.4995 | 0.2003 | 0.2685 | 0.1136 |
| 6 | 0.4998 | 0.1933 | 0.2633 | 0.1097 |
| 7 | 0.4959 | 0.1867 | 0.2605 | 0.1076 |
| 8 | 0.5018 | 0.1568 | 0.2740 | 0.1178 |
| 9 | 0.4879 | 0.1393 | 0.2670 | 0.1126 |
| 10 | 0.4902 | 0.1582 | 0.2692 | 0.1142 |
| Mean | 0.5000 | 0.1745 | 0.2666 | 0.1123 |
| SD | 0.0079 | 0.0192 | 0.0039 | 0.0029 |
| 95% CI | [0.4951–0.5049] | [0.1626–0.1864] | [0.2642–0.2690] | [0.1105–0.1141] |

random seeds when splitting the data. The observed small variations in the model's performance across these trials serve as a valuable indication of its high stability. The consistency demonstrated by the model across multiple iterations provides reassurance that its performance is not overly influenced by random fluctuations. This stability is a desirable characteristic, as it suggests that the model's performance can be reliably reproduced in different settings and scenarios.

## CONCLUSIONS

In this study, we introduced an attention-based bidirectional long short-term memory network as a novel approach for Chinese relation extraction. The experimental results presented in this research clearly indicate the effectiveness of our proposed method in addressing the challenges associated with relation extraction in the Chinese language. The incorporation of the attention mechanism significantly enhances the extraction efficiency by allowing the model to focus on relevant features and capture important contextual information. Further advancements and optimizations in this field can build upon our research, paving the way for more accurate and efficient relation extraction techniques in Chinese natural language processing tasks.

### Funding

This work is supported by the Language Ecology Research and Language Resource Database Construction in The South China Sea Project (Grant number: 25016207). The

funders had no role in study design, data collection and analysis, decision to publish, or preparation of the manuscript.

## Grant Disclosures

The following grant information was disclosed by the authors:
The South China Sea Project: 25016207.

## Competing Interests

The authors declare that they have no competing interests.

## Author Contributions

- Yanzi Zhang conceived and designed the experiments, performed the experiments, analyzed the data, performed the computation work, prepared figures and/or tables, authored or reviewed drafts of the article, and approved the final draft.

## Data Availability

The code and data are available in the Supplemental File.

The datasets used in this study were retrieved from Li et al. (2019) and are available at GitHub: https://github.com/thunlp/Chinese_NRE/tree/master/data.

Li, Z., Ding, N., Liu, Z., Zheng, H., and Shen, Y. (2019). Chinese relation extraction with multi-grained information and external linguistic knowledge. In: Proceedings of the 57th Annual Meeting of the Association for Computational Linguistics. Association for Computational Linguistics.

## Supplemental Information

Supplemental information for this article can be found online at http://dx.doi.org/10.7717/peerj-cs.1509#supplemental-information.

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
