# Peer review of "Relation extraction in Chinese using attention-based bidirectional long short-term memory networks"

_PeerJ Computer Science, doi:10.7717/peerj-cs.1509_

## Round 0.1 · original submission · Minor Revisions

It is necessary to revise the manuscript in order to incorporate the reviewers' comments. Specifically, the comments of Reviewer 2 indicate that the unclear aspects of the experiments should be clarified. As suggested by Reviewer 1, the confidence interval for the performance of both models must also be computed.

Reviewer 1 ·

Basic reporting

The manuscript is well-organized with professional language used. The background is relatively sufficient but it needs to be updated with several recent studies. Figures and tables are clearly presented. However, there are several questions remain.
(1) Is there any other benchmark datasets besides, SanWen and FinRE?
(2) Did you split the training, validation, and test set or they are already split?
(3) More information on original LSTM needs to be added.
(4) Section about "Attention" needs to be enriched with more detailed information.

Experimental design

The research fails with the Aims and Scope of the journal with well defined research questions. The model architecture is clearly explained with details. The methods are described with adequate information. However, more details on "Attention" sections are required to added.

Validity of the findings

The experiments were repeated 10 times to provide mean and standard deviation. Please calculate Confidence interval for both models of these two datasets. The conclusion is clear and in accordance with research question and experiments.

Reviewer 2 ·

Basic reporting

- According to my assessment, the article has several positive aspects. The English writing is clear and unambiguous. The inclusion of adequate references and background information also strengthens the article. Furthermore, providing raw data and code improves the article's transparency and reproducibility.
- In terms of structure, the article is well-organized, allowing for a logical flow of ideas. Despite the fact that there is only one figure, its quality is good, effectively supporting the information presented.
- The manuscript can be considered self-contained as it provides relevant results that contribute to the overall understanding of the subject matter.

Experimental design

- The manuscript demonstrates an application of attention-based bidirectional long-short-term memory networks, a relatively recent neural network architecture. This exemplifies the authors' innovative use of cutting-edge techniques.
- There are no apparent indications of information leakage or overfitting during the modeling process, suggesting that the methodology has been meticulously implemented. This demonstrates the authors' meticulous attention to detail and dedication to ensuring the accuracy of their findings.
- The inclusion of code with the article is another point, as it increases the reproducibility of the results. By releasing the code, other researchers are able to validate and expand upon the findings, fostering transparency and advancing the field.

Validity of the findings

- The findings are useful in practice and the results are statistically sound.
- Conclusions are well stated and linked to the original research question.

Additional comments

- Please clarify this a particular point. The author mentioned performing 10 trials of the experiment using random seeds for data splitting. Could you please specify which sets of the data were modified by this process? Were modifications made to all three sets (training, validation, and test) or just the training and validation sets? In addition, it would be useful to know which trial the numerical information in the Experiments section, such as training time and epochs when the model converged, pertains to.
- In addition, there are some minor errors in the References: "chinese" in Chen et al. (2014), Peng et al. (2016), and Zhang et al. (2017), and "lstm" in Geng et al. (2020) should be capitalized.

---

## Round 0.2 · accepted · Accept

I have assessed the revision and I am happy with the revised manuscript. The authors have already addressed all the (minor) comments from the reviewers. It is not necessary to send this version to the reviewers. The current version should be ready for publication.